# Immune and Genomic Analysis of Boxer Dog Breed and Its Relationship with *Leishmania infantum* Infection

**DOI:** 10.3390/vetsci9110608

**Published:** 2022-11-02

**Authors:** Luis Álvarez, Pablo-Jesús Marín-García, Pilar Rentero-Garrido, Lola Llobat

**Affiliations:** 1Departamento Producción y Sanidad Animal, Salud Pública y Ciencia y Tecnología de los Alimentos, Facultad de Veterinaria, Universidad Cardenal Herrera-CEU, CEU Universities, 46010 Valencia, Spain; 2Precision Medicine Unit, INCLIVA Biomedical Research Institute, 46010 Valencia, Spain

**Keywords:** canine breed, leishmaniosis, boxer dog, *Leishmania*, Mediterranean region

## Abstract

**Simple Summary:**

Leishmaniosis is a zoonotic disease, endemic in 88 countries, including those from the Mediterranean region. Several authors indicate differences in susceptibility and resistance to leishmaniosis in different canine breeds, with boxer being one of the breeds with a higher prevalence of the disease. This study analyzes the serum profiles of cytokines related to the immune response, together with the screening of genomic variants fixed in boxer breed samples, to understand their differential susceptibility to *L. infantum* infection. The results of this study indicate new pathways related to *L. infantum* infection and immune response in boxers, involving genes related to interleukin and toll-like receptors, as well as to the immune system and the regulation of expression. Future studies are required to elucidate the role of specific genes in the *L. infantum* infection mechanism in this canine breed.

**Abstract:**

Leishmaniosis, one of the most important zoonoses in Europe, is caused *by Leishmania infantum*, an intracellular protozoan parasite. This disease is endemic in the Mediterranean area, where the main reservoir is the dog. Several studies indicate a possible susceptibility to *L. infantum* infection with clinical signs in some canine breeds. One of them is the boxer breed, which shows a high prevalence of disease. In this study, immunological and genomic characterization of serum samples from boxer dogs living in the Mediterranean area were evaluated to analyze the immune response and the possible genetic explanation for this susceptibility. Serum levels of cytokines IFN-γ, IL-2, IL-6, IL-8, and IL-18 were determined by ELISA commercial tests, while the genotyping study was performed using the CanineHD DNA Analysis BeadChip. The results show relevant differences in the serum levels of cytokines compared to published data on other canine breeds, as well as sequence changes that could explain the high susceptibility of the boxer breed to the disease. Concretely, polymorphic variants in the *CIITA*, *HSF2BP*, *LTBP1*, *MITF*, *NOXA1*, *PKIB*, *RAB38*, *RASEF*, *TLE1*, and *TLR4* genes were found, which could explain the susceptibility of this breed to *L. infantum* infection.

## 1. Introduction

The boxer, or Deutscher boxer, is a canine breed officially recognized by The Kennel Club [1]. It was firstly internationally recognized in 1955 by the “Federation Cynologique Internationale” (FCI-AISBL) with the breed standard number 144, classified in group 2—Section 1 (Mollosoid breeds, mastiff type—with working trial), and the breed had an official valid standard from 2008 [2]. As reported by The Kennel club and FCI, the boxer is a descendant of the Bullenbeisser, a German breed, which was used to hunt bear, boar, and deer until the 19th century. According to The Kennel Club, the physical characteristics of boxer dogs are height of 22.5–25 and 21–23 inches in males and females, respectively, and weight between 66–70 (males) to 55–60 (females) pounds, and the mean of life expectancy is around 10 years [3] (Figure 1).

This canine breed seems to be particularly resistant to several metabolic diseases, such as diabetes mellitus [4], whereas it presents a higher prevalence of other diseases, such as congenital heart disease (12.86% prevalence) [5], genetic heart disease as the familiar ventricular arrhythmias, an autosomal dominant condition with varied penetrance inheritance [6], granulomatous colitis, a severe form of inflammatory bowel disease [7], and hematopoietic diseases such as non-Hodgkin lymphoma [8] or T-cell lymphomas [9]. This last has been linked in boxers with Phosphate and Tensin Homolog (*PTEN*) and Special AT-rich Sequence Binding protein 1 (*SATB1*) gene mutations [10,11]. These genes are also related to immune response, as *PTEN* is a relevant protein in the maintenance of immune homeostasis, antiviral and T-cell response [12,13], and *SATB1* is a chromatic organizer essential for controlling genes participating in T-cell development and activation [14] and essential for Th2 cytokine expression [15]. Multiple studies related these genes with immune response to parasitic infections, including *Echinococcus granulosus*, *Trypanosoma cruzi*, *Schistosoma japonicum*, and *Leishmania* spp. [16,17,18,19,20,21,22]. In fact, several authors cited a higher prevalence of *Leishmania* spp. infection in boxer than in other canine breeds [23,24]. Leishmaniosis is a parasitic disease caused by different genera of *Leishmania*, including *L. infantum.* The most common clinical manifestations of canine leishmaniosis are generalized lymphadenomegaly, loss of body weight, decreased or increased appetite, lethargy, mucous membrane pallor, splenomegaly, polyuria and polydipsia, fever, vomiting and diarrhea [25]. This last is the most prevalent causal agent of leishmaniosis infection in the Mediterranean region, which is transmitted by the sandfly *Phlebotomus perniciosus*. This zoonotic disease is endemic in 88 countries and is considered the most relevant vector-borne disease in the Mediterranean region (Figure 2), affecting between 63 and 80% of the domestic dog population [25,26,27].

The global seroprevalence in domestic dogs in Spain was estimated at 10.12% [29], whereas in the boxer breed, the prevalence reached 39.13%, being one of the canine breeds most affected [23]. Genomic analysis to date pointed out that several polymorphisms could be related to resistance or susceptibility to canine leishmaniosis. Four single nucleotide polymorphisms (SNPs) in the canine β-defensin-1 (CBD1) gene have been associated with *L. infantum* infection in dogs from Italy [30]. Two polymorphisms in the canine Slc11a1 gene, one intronic single SNP and one silent SNP in exon 8 were associated with an increased risk of canine visceral leishmaniosis [31]. Recently, a meta-analysis related some genetic variants of this gene with the infectious disease response in humans [32] and, concretely, with leishmaniosis [33]. In boxers, the haplotype TAG-8-141 in the Slc11a1 gene has been associated with this disease [34].

This work carried out an immunological and genetic characterization of boxer breed samples to elucidate the underlying mechanisms of immune response related to *L. infantum* infection.

## 2. Materials and Methods

### 2.1. Ethical Approval

The experiments involving animals were conducted according to the guidelines of the Declaration of Helsinki and approved by the Animal Experimentation Ethics Committee of the Universidad Cardenal Herrera CEU, with code 2020/VSC/PEA/0216.

### 2.2. Animals and Epidemiological Data

A total of 31 serum samples of boxer breed dogs living in the Valencia Community region (Eastern Spain, Mediterranean region) were obtained for the study, along with epidemiological data from October 2021 to May 2022. The following epidemiological data were recovered for all animals: sex (male or female), age (puppies—less than 1 year old, young—between 1 and 5 years old, adults—between 5 and 10 years old, and elder—more than 10 years old), official vaccination status, leishmaniosis vaccination status, external deworming status and type, living conditions, type of feeding, and *L. infantum* infection status. If the infection was present, the clinical manifestation status was also collected.

### 2.3. Samples Collection and Cytokine Levels

Ten milliliters of whole blood were drawn by cephalic venipuncture with Vacutainer tubes without anticoagulant. Samples were maintained at room temperature to obtain serum aliquots, which were stored at −20 °C until processing. Serological testing for detection of specific *L. infantum* antibodies was performed using the indirect immunofluorescent antibody test (IFAT) for anti-*Leishmania* specific immunoglobulin G (IgG) antibodies (MegaFLUO^®^ LEISH, Megacor Diagnostik GmbH, Hörbranz, Austria). Samples were considered seropositive with IFAT titer ≥ 1/80, following the manufacturer’s instructions [35]. The whole blood samples were used for DNA extraction before 24 h to recovery.

The IL-2, IL-6, IL-8, IFN-γ, (Canine IL-2 ELISA kit, Canine IL-6 ELISA kit, Canine IL-8 ELISA kit, and Canine IFN-γ ELISA kit, Invitrogen, respectively), and IL-18 (Canine IL-18 ELISA kit, Mybiosource) levels were measured in serum samples by a commercial ELISA kit method following the manufacturer’s recommendations. In brief, 50 µL of serum was used for the analysis with sandwich ELISA. The microplate was pre-coated with an antibody specific to cytokines. Samples were added to the microplate wells and combined with the specific antibody. Then, a biotinylated detection antibody specific to each cytokine and avidin-horseradish peroxidase (HRP) conjugate were added successively to each microplate well and incubated. Free components were washed away. The substrate solution was added to each well. The enzyme–substrate reaction was determined by the optical density (OD) and measured spectrophotometrically at a wavelength of 450 nm in a Victor-X3™ plate reader (Perkin Elmer^®^, Waltham, MA, USA). The concentration of each cytokine was calculated by comparing the OD of the samples to the standard curve.

### 2.4. DNA Extraction and Whole Genome Analysis

Genomic DNA (gDNA) from 24 boxer samples was isolated using a QIAamp DNA Blood Kit following the manufacturer’s protocol (QIAamp; Qiagen, Hilden, Germany). DNA was quantified using the Glomax^®^ Discover Fluorimeter (Promega, Madison, WI, USA) and the QuantiFluor^®^ dsDNA Kit (Promega, Madison, WI, USA). gDNA concentrations for all samples were a minimum of 50 ng/µL. DNA samples were whole-genome amplified for 20–24 h at 37 °C, fragmented, precipitated and resuspended in an appropriate hybridization buffer.

Samples were genotyped using the CanineHD DNA Analysis BeadChip WG-440-1001 (Illumina, Inc., San Diego, CA, USA) and hybridized on the prepared BeadChips for 16–24 h at 48 °C. Following the hybridization, nonspecifically hybridized samples were removed by washing, while the remaining specifically hybridized loci were processed for the single-base extension reaction, stained, and imaged on an Illumina iScan Reader. GenomeStudio 2.0.5 (Illumina Inc., San Diego, CA, USA) was used to process data generated from the iScan system for subsequent analysis, according to manufacturer guidelines. Intensity data were loaded into the Genotyping Module for primary data analysis, including raw data normalization, clustering and genotype calling.

SNPs on sex chromosomes and with a call rate <95% were discarded using PLINK v1.90b6.22 software [36,37,38,39]. The final data set included 165,480 mapped positions in 24 boxer samples with a mean genotyping rate of 0.988.

### 2.5. Analysis of Polymorphisms Related to Immune Response

The allelic status of previously published polymorphisms and genomic regions described to be related to the immune response to *L. infantum* infection in dogs [31,40,41,42,43,44,45,46,47] was interrogated in the analyzed samples.

PLINK v1.90b6.22 software was used to extract variants from the selected genome regions, encompassing 38 genes, according to the mapping information of the *Canis lupus familiaris* genome assembly CanFam3.1., and to calculate allele frequencies. Polymorphisms were considered as fixed in the boxer samples when presenting frequencies above 0.7. Detected polymorphisms were annotated using NCBI refseqs, release 105. The rsID information was downloaded and annotated from the European Variation Archive EVA release 3 corresponding to the CanFam3.1 assembly.

## 3. Results

Of the 31 samples, 17 were males and mainly adult (14) or young (10), and 28 were fed with commercial feed (Table 1).

Only one of the analyzed dogs did not present the regulatory vaccination, and the majority were not vaccinated against *Leishmania* (83.87%). Related to lifestyle, only two dogs did not live with other animals, and all of the animals were kept outdoors and used external deworming. Likewise, one boxer presented positive results for antibodies against *Leishmania* with the IFAT technique (1/320), whereas two presented a 1/80 titer. None of the animals presented clinical signs during the study. Serum cytokine levels were analyzed. The concentration values of analyzed cytokines were 0.22 ± 0.14 ng/mL for IFN-γ, 68.57 ± 12.09 ng/mL for IL-2, 0.62 ± 0.23 ng/mL for IL-6, 263.75 ± 152.73 pg/mL for IL-8, and 43.08 ± 7.09 ng/mL for IL-18 (Table 2).

The genomic data did not show differential variants in the analyzed cytokine encoding genes. However, several genes presented SNPs fixed in the boxer population analyzed. Concretely, 20 intronic variants were found in Latent Transforming Growth Factor Beta Binding Protein 1 (*LTBP1*), Class II Major Histocompatibility Complex Transactivator (*CIITA*), genes encoding kinases such as CAMP-dependent Protein Kinase Inhibitor Beta (*PKIB*), Heat Shock Transcription Factor 2 Binding Protein (*HSF2BP*) and members of theRAS Family such as RAS And EF-Hand Domain-Containing Protein (*RASEF*) and *RAB38*. Melanocyte Inducing Transcription Factor (*MITF*) gene presented a downstream variant that contained a non-coding transcription exon (concretely, lncRNA). Six intronic variants were found in TLE Family Member 1, Transcriptional Corepressor (*TLE1*), an expression gene regulator, and one intronic variant in NADPH Oxidase Activator 1 (*NOXA1*). Finally, gene encoding Toll-Like Receptor 4 (*TLR4)* showed a downstream and 5′UTR variant (Table 3). Other SNPs with frequencies lower than 0.7 in the same or in other genes are summarized in Appendix A.

## 4. Discussion

The boxer is a canine breed recognized internationally in 1955 by the “Federation Cynologique Internationale” (FCI-AISBL). The results of the present paper show the immune and genomic profile of this canine breed. Concretely, the concentration values for the analyzed cytokines were 0.22 ± 0.14 ng/mL for IFN-γ, 68.57 ± 12.09 ng/mL for IL-2, 0.62 ± 0.23 ng/mL for IL-6, 263.75 ± 152.73 pg/mL for IL-8, and 43.08 ± 7.09 ng/mL for IL-18. However, other authors found serum levels of these cytokines that were different from these results, being around 0.5 ng/mL and 0.3 ng/mL for IFN-γ in Ibizan hound dogs and crossbreed, respectively, around 60 ng/mL for IL-2, and around 180 pg/mL and 200 pg/mL for IL-8 in Ibizan hound and crossbreed, respectively [48]. Other authors showed levels of IFN-γ around 30 µg/mL [49], and 17 pg/mL, 137 pg/mL, and 83 pg/mL for IL-2, IL-8, and IL-18, respectively [50].

This canine breed is one of the breeds with the highest prevalence of *Leishmania* infection [23,51], and the immunological profile of this canine breed showed relevant differences in several cytokine levels compared to those published in other canine breeds. In fact, previous studies indicated lower levels of IL-2 and IFN-ϒ in the boxer breed compared to other canine purebreds or crossbreed [48]. The ability of the host to control *Leishmania* infection requires a strong cellular immune response, associated with the activation of T helper (Th)-1 cells producing IL-2 [52,53]. Some studies of IL-2 and IFN-ϒ noticed their relationship with a control and protective response against *Leishmania* infection by the host [54,55,56]. None of genes encodes the cytokines analyzed, nor have other, related cytokines been found, which indicates that these different serum profiles of this canine breed are the result of more complex genetic mechanisms and expression regulation.

In fact, 20 intronic polymorphisms have been found in *LTBP1*, a gene known to be involved in T regulatory lymphocyte differentiation [40]. In fact, Batista et al. (2016), in a genome-wide association study of cell-mediated response in dogs infected by *L. infantum,* showed several genetic markers that could explain the phenotypic variance in the cytokines’ expression, including *LTBP1* [40]. The levels of this pleiotropic cytokine are positively correlated with the expression of TGF-β [57] and with other immune functions, including regulation of adaptative immune response, T-cell selection, and Th1- and Th2- cell differentiation [55]. Moreover, this cytokine promotes IL-7R expression [58], plays a key role in Treg cell development [59], and regulates the Notch pathway, among others (see review [60]).

An intronic change, 6:31796528A > G, has been observed in the *CIITA* gene. This intronic variant, which overlaps six different transcripts, does not have a known biological function. However, *CIITA* is a key regulator of *IL6* and *HLAII* expression [61,62], and epigenetic regulation of this gene by the acetylation of *CIITA* promoters has been observed after *Leishmania* parasitic infection [63]. Previous studies demonstrated that the downregulation of *CIITA* expression appeared after the infection of macrophages with *Mycobacterium bovis.* The infection is controlled by the *Nramp1* gene [64], which is related to the susceptibility against visceral leishmaniosis [31,65]. Other genes related to immune response also present intronic variants, such as *PKIB*, the kinase activator of PI3/Akt pathway [66], which mediates different receptors including cytokine receptors and regulates macrophage response [67]. The *RASEF* gene, which showed five SNPs, one of them being a missense variant, and *RAB38* (one intronic variant) have not been directly related to the immune response, although other members of the Ras family have been associated with different immune pathways. For example, N-Ras, K-Ras and H-Ras regulate IL-10 and IL-12 production by CD40 and ERK-1/2 pathways [68]. In fact, Chakraborty et al. (2015) demonstrated the relationship of N-Ras and K-Ras in *L. major* infection and its regulation by Toll-Like Receptor 2 (TLR-2) [69]. Another toll-like receptor, TLR-4, showed a 5′UTR variant in this study. *TLR2* expression was higher in dogs naturally infected by *L. infantum* than in seronegative healthy dogs, whereas *TLR4* did not present differences [70]. However, several studies indicate that production of IL12 by macrophages is TLR-4 dependent in *L. mexicana* infection [71]. These results, together with those observed in our study, indicate that future studies analyzing the relationship between TLR-4, IL-12 and its receptor are necessary in relation to *L. infantum* and the disease’s progression.

So far, there is no evidence of a direct relationship between *HSF2BP* and the immune response, although its functions related to the regulation of cell adhesion, autophagy response and constitutive transcription regulation are well known [72,73]. Other family members of HSF have been related to immunity. HSP60 regulates the expression of Th1/Th2 transcription factors in human cells in vitro, leading to decreased secretion of IFN-ϒ and increased secretion of IL-10 [74]. Finally, HSP72 activates neutrophil functions via the PI3K/ERK pathway with TLR-2 participation [75]. For its part, *TLE1*, which showed six intronic variants, has been related to NF-κB pathway regulation [76]. Both canonical and non-canonical NF-κB pathways have been related to innate immune response regulation [77,78]. In this regard, Mitchell et al. (2016) published a comprehensive review, explaining these two pathways and related molecules, including toll-like receptors and several cytokines [79]. Utsunomiya et al. (2015) found an associated marker for visceral leishmaniosis in chromosome 1, located between two predicted transcription factor binding sites regulating the expression of *TLE1*, a key regulator of Notch signaling, which regulates macrophage activity and Th1/Th2 differentiation [45]. Other genes related to innate immune response are *NOXA1* [80] and *MITF* [81]. The last one showed a downstream variant (non-coding transcription exon) in our study. *MITF* regulates several genes in response to infection by some pathogens, such as *Vibrio parahaemolyticus* [82], and is related to activation of B lymphocytes in *L. longipalpis* infection [42].

None of the variants found in this study resulted in a change in amino acids, since all of were in intronic or regulatory regions 3’UTR or 5’UTR. These results indicate that, probably, the defective immune response in *Leishmania* infection of the boxer canine breed is more complex and includes different regulation factors and pathways, including epigenetic mechanisms and expression regulation genes.

## 5. Conclusions

The boxer, or Deutscher boxer, is a canine breed officially recognized by the Kennel Club and internationally in 1955 by the FCI-AISBL, with susceptibility to *L. infantum* infection. This descriptive study showed levels, in boxers, of some cytokines that were lower than published data for other canine breeds. The genomic analysis revealed many variants that were fixed in this dog population in genes related to immune response regulation. The most relevant variants found were in *MITF*, which shows a downstream variant with a non-coding transcription exon, and were related to B lymphocyte activation, and in *TLR4*, which regulates IL-12 production in *L. mexicana* infection. These results indicate that this breed presents specific variants in genes related to regulation of the immune system and its response to *L. infantum* infection, which could explain the high prevalence of leishmaniosis found in boxer dogs. Further studies are necessary to elucidate the relationship between these identified variants and immune response against infection by *Leishmania*.

## Figures and Tables

**Figure 1 vetsci-09-00608-f001:**
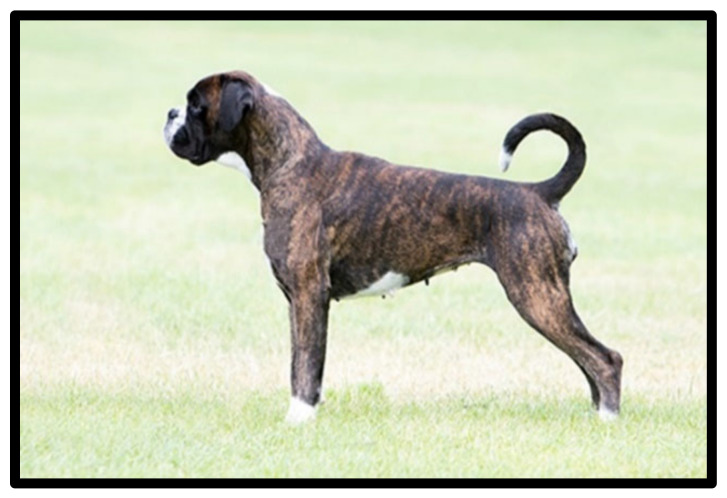
Photograph of adult male of boxer breed (extracted from The Kennel club materials).

**Figure 2 vetsci-09-00608-f002:**
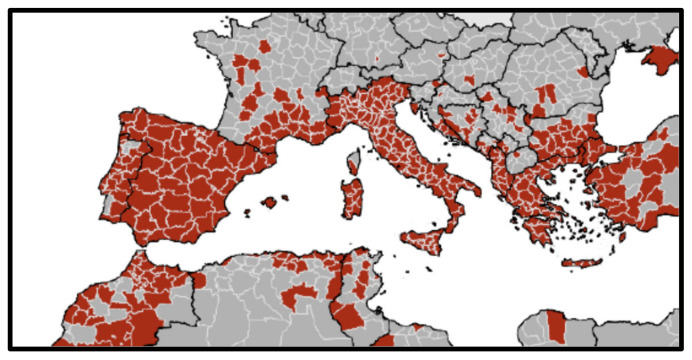
Geographical distribution of reported human and/or animal cases of leishmaniosis due to *L. infantum* between 2009 to 2020 (data obtained by Technical Report of European Centre for Disease Prevention and Control (ECDPC) [28]).

**Table 1 vetsci-09-00608-t001:** Epidemiological data recovered for the 31 boxer samples analyzed.

Variable	Categories	No. of Dogs (%)
Gender	Male	17 (54.84)
	Female	14 (45.16)
Age	Puppy (<1 year)	4 (12.90)
	Young (1 to 5 years)	10 (32.26)
	Adult (5 to 10 years)	14 (45.16)
	Elder (>10 years)	3 (9.68)
Diet	Commercial	28 (90.32)
	Home prepared/raw food consumption	3 (9.68)
Overall		31 (100.00)

**Table 2 vetsci-09-00608-t002:** Concentrations of the analyzed cytokines. The table shows the number of animals (*n*), range, mean ± standard deviation (SD), and coefficient of variation (CV).

Cytokine ^1^	*n*	Range ^2^	Mean ± SD ^2^	CV (%)
IFN-γ	31	0.01–0.78	0.22 ± 0.14	62.29
IL-2	31	10.37–721.13	68.57 ± 12.09	17.63
IL-6	31	0.40–1.39	0.62 ± 0.23	37.10
IL-8	31	58.48–624.40	263.75 ± 152.73	57.91
IL-18	31	0–353.45	43.08 ± 7.09	16.46

^1^ IFN-γ: interferon gamma, IL: interleukin. ^2^ The values for IL-8 are expressed in pg/mL, and for IFN-γ, IL-2, IL-6 and IL-18, in ng/mL.

**Table 3 vetsci-09-00608-t003:** Genomic variants, found in genes other than the cytokines analyzed, with a frequency of the alternative allele above 0.7. Table showing the identify number (rsID), chromosome position, reference and alternative alleles, frequency of alternative variant in the analyzed dataset, and functional class of variant.

Gene ^1^	rsID	Chromosome Position	Ref. Alt. ^2^	Frequency	Functional Class of Variant
*CIITA*	rs24353887	6:31796528	A-G	0.9792	Intronic
*HSF2BP*	rs23697150	31:37627924	A-G	0.7917	Intronic
*LTBP1*	rs22564606	17:26144284	C-T	0.9792	Intronic
	rs22598434	17:26182161	A-G	0.7609	Intronic
	rs22598480	17:26192010	C-T	0.8478	Intronic
	rs22598531	17:26217715	T-A	0.7292	Intronic
	rs22598552	17:26233826	G-T	0.7609	Intronic
	rs22583570	17:26237829	A-G	0.7292	Intronic
	rs22583641	17:26261335	A-G	0.7391	Intronic
	rs22583674	17:26274101	C-T	0.9583	Intronic
	rs22583693	17:26283669	G-A	0.8750	Intronic
	rs22573465	17:26292549	A-G	0.8958	Intronic
	rs22583733	17:26328978	T-C	0.9583	Intronic
	rs22583751	17:26345347	G-A	0.7292	Intronic
	rs22617468	17:26365885	C-A	0.8125	Intronic
	rs22617490	17:26390401	C-T	0.8261	Intronic
	rs22565050	17:26412722	G-A	0.7292	Intronic
	rs22565078	17:26425762	C-G	0.7292	Intronic
	rs22565091	17:26437220	A-G	0.8696	Intronic
	rs22585869	17:26451018	C-G	0.7391	Intronic
	rs22585928	17:26456566	A-G	0.7391	Intronic
	rs22600112	17:26478030	G-A	0.7500	Intronic
*MITF*	rs8519356	20:21871904	T-C	0.7500	Downstream
*NOXA1*	rs24534859	9:48312935	G-T	0.8125	Intronic
*PKIB*	rs21974900	1:62026582	C(G)-A	0.9792	Intronic
*RAB38*	rs22921195	21:12120865	T-C	0.7174	Intronic
*RASEF*	rs21892604	1: 76327161	G-A	0.7174	Intergenic
	rs21885698	1: 76366341	C-T	0.7174	Intergenic
	rs21894538	1: 76416494	C-T	0.7174	Intronic
	rs21913661	1: 76423140	A-G	0.7174	Intronic
	rs21984010	1: 76452566	G-A	0.7174	Missense
*TLE1*	rs852602083	1:77556658	G-T	1	Intronic
	rs22038874	1:77569697	G-C	1	Intronic
	rs22038878	1:77576847	T-C	1	Intronic
	rs21881897	1:77607385	T-A	0.9783	Intronic
	rs22038945	1:77612762	A-G	0.9783	Intronic
	rs22038982	1:77625675	C-T	1	Intronic
*TLR4*	rs22145736	11:71364581	T-C	0.7500	5′UTR

^1^*CIITA*: Class II Major Histocompatibility Complex Activator; *HSF2BP:* Heat Shock Transcription Factor 2 Binding Protein; *LTBP1*: Latent Transforming Growth Factor Beta Binding Protein 1; *MITF:* Melanocyte Inducing Transcription Factor; *NOXA1*: NADPH Oxidase Activator 1; *PKIB*: CAMP-Dependent Protein Kinase Inhibitor Beta; *RAB38*: RAB38 Member RAS Oncogene Family; *RASEF*: RAS And EF-Hand Domain-Containing Protein; *TLE1*: TLE Family Member 1, Transcriptional Corepressor; *TLR4*: Toll-Like Receptor 4. ^2^ C: cytosine, T: thymine; A: adenine; G: guanine.

## Data Availability

Not applicable.

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
