# Peer review of "Immune and Genomic Analysis of Boxer Dog Breed and Its Relationship with Leishmania infantum Infection"

_vetsci, 2022, doi:10.3390/vetsci9110608_

Round 1
Reviewer 1 Report
The study authors conclude that "The results show relevant differences in the serum levels of cytokines compared to published data in other canine breeds, as well as sequence changes that could explain the high susceptibility to the disease of the boxer breed. Concretely, polymorphic variants in the CIITA, HSF2BP, LTBP1, MITF, NOXA1, PKIB, RAB38, 32 RASEF, TLE1, and TLR4 genes have been found, which could explain susceptibility of this breed to L. infantum infection".
My concern is that the data presented may not be sufficient to support the conclusion, for example, the authors did not provide the sera level form the control (non Boxer) dogs to compare the sera levels.
yes, the genetic variation is noted based on the sequence analysis.
The authors should present any supplemental data to support he conclusion. The two tables presented as supplemental and non-publishes are the same.
The authors are very careless in their presentation. Half of the manuscript is in red and other half is in black letters and the Table is completely messed up. All these should be corrected for review again.
Best wishes,
Author Response
Many thanks to the reviewer for his comments and suggestions. We have answered point by point in the attached document.

Reviewer 2 Report
The authors show levels of 5 cytokines in boxers and describe some polymorphisms that appear to be fixed in boxers. This data by itself probably does not warrant publication.
The authors try to relate their findings to the increased prevalence of leishmaniosis in boxers as compared to other breeds. However, there is no direct demonstration that the variants they identified are related to a presumed increased susceptibility of boxers to leishmaniosis. They do cite the literature, but it is conjecture about possible roles in immune regulation.
There is no information in regards to the increased susceptibility of boxers to leishmaniosis being genetic. Could the increase prevalence have a behavioral or environmental component?
The authors state that the cytokine levels are different in boxers, but do not show this data and only reference other papers. The values from these other papers should also be included in Table 2 so the readers can evaluate the degree of difference and whether it is biologically significant.
The genetic variants identified are almost all in non-coding regions. Furthermore, most are in introns. Therefore, these variants are unlikely to affect the function of the gene nor the expression of the gene. This means the variants are extremely unlikely to be involved in the increased susceptibility to leishmaniosis.
Author Response

(The authors gave the same response as above.)

Reviewer 3 Report
The study author by Luis Alvarez et al. investigated the immune and gene involvement of L. infantum infected boxer dog breeds. The methods and discussion part were written clearly with citations. However, the authors need to clarify few points for the improvement of the article.
Major Comments
1. Title: The authors need to change or modify the word "characterization" to "identification/analysis/representation"?
2. Introduction: Explain what are the symptoms of the L. infantum infection in boxer dogs?
3. Methods-part 2.5: Why the authors not used to map the recent dog genome assembly like ROC-CanFam 1.0 or not performed genome mapping against boxer breed genome to assemble (Dog10K-boxer_Tasha)? CanFam 3.1 is not a recent genome assembly deposited in the genome data viewer.
4. Results: Provide the selected gene sequences with gene/rsID in the supplementary file.
5. Table-2: The results are not clearly explained. Explain the concentration values of the analyzed cytokines in the result part.
6. Discussion: Rewrite sentence Line 261-263. Is that the present results recognized by the FCI-AISBL?
7. Line 310-311: Not relevant to the present results and species discussing in this article.
Minor comments
1. Follow the unique word all over the manuscript: Leishmaniasis or Leishmaniosis? Line 87 used Leishmaniasis?
2. Line 76: Expansion: ECDPC?
3. Line 163: What are the 3 files corresponding to the CanFam 3.1 assembly?
4. Repeated sentence: Line 334-335: in the introduction and line 261-262 in discussion. Rewrite sentence.
Author Response

(The authors gave the same response as above.)

Reviewer 4 Report
Leishmaniasis is a zoonotic disease caused by protozoan parasites of the genus Leishmania and transmitted by phlebotomine sandflies. The disease is one of the neglected tropical diseases (NTDs), endemic in 88 countries and/or regions, affecting humans and different wild and domestic mammals (reservoir hosts), including canines.
In this zoonosis, it has been demonstrated that the Deutscher boxer seems to present susceptibility against Leishmania (Leishmania) infantum infection, with higher prevalence of leishmaniasis than other canine breeds, showing low levels of some cytokines than other canine breeds. Here, the current study performed an immunological and genetic characterization of boxer dog breed and its relationship with the Leishmania infection, analyzing 31 sera samples of boxer breed dogs from Valencia, Eastern Spain. As mentioned by the authors, the results obtained indicate new pathways related to L. (L.) infantum infection and immune response in boxers, involving genes related to interleukin and toll-like receptors, as well as to the immune system and the expression regulation. However, the role of specific genes in the Leishmania infection mechanism(s) remains to be elucidated precisely in future.
The study is well designed and organized, and the manuscript is well written; besides, it is better to consider the English language in terms of comprehension and fluency.
Minor comments/suggestions:
The usage of Leishmania (Leishmania) infantum (= L. (L.) infantum) throughout the text is recommended.
Table 2. Replace "The table shows" by ; showing....
Table 3. Replace "Table shows" by ; showing....
Author Response

(The authors gave the same response as above.)
